# MicroRNA-10 Family Promotes Renal Fibrosis through the VASH-1/Smad3 Pathway

**DOI:** 10.3390/ijms25105232

**Published:** 2024-05-11

**Authors:** Yichen Shuai, Na Xu, Chuan Zhao, Fengrui Yang, Zhifen Ning, Guoxia Li

**Affiliations:** Department of Pharmacology, School of Basic Medical Sciences, Tianjin Medical University, Tianjin 300070, China

**Keywords:** renal fibrosis, MicroRNA-10, Vasohibin-1, Smad3

## Abstract

Renal fibrosis (RF) stands as a pivotal pathological process in the advanced stages of chronic kidney disease (CKD), and impeding its progression is paramount for delaying the advancement of CKD. The miR-10 family, inclusive of miR-10a and miR-10b, has been implicated in the development of various fibrotic diseases. Nevertheless, the precise role of miR-10 in the development of RF remains enigmatic. In this study, we utilized both an in vivo model involving unilateral ureteral obstruction (UUO) in mice and an in vitro model employing TGF-β1 stimulation in HK-2 cells to unravel the mechanism underlying the involvement of miR-10a/b in RF. The findings revealed heightened expression of miR-10a and miR-10b in the kidneys of UUO mice, accompanied by a substantial increase in p-Smad3 and renal fibrosis-related proteins. Conversely, the deletion of these two genes led to a notable reduction in p-Smad3 levels and the alleviation of RF in mouse kidneys. In the in vitro model of TGF-β1-stimulated HK-2 cells, the co-overexpression of miR-10a and miR-10b fostered the phosphorylation of Smad3 and RF, while the inhibition of miR-10a and miR-10b resulted in a decrease in p-Smad3 levels and RF. Further research revealed that miR-10a and miR-10b, through binding to the 3’UTR region of Vasohibin-1 (VASH-1), suppressed the expression of VASH-1, thereby promoting the elevation of p-Smad3 and exacerbating the progression of RF. The miR-10 family may play a pivotal role in RF.

## 1. Introduction

Chronic kidney disease (CKD) is a progressive and irreversible impairment of kidney structure and function that may eventually result in renal failure. Renal fibrosis (RF) emerges as a key histological feature in the advancement of CKD, characterized by the gradual replacement of normal renal tissue with fibrous tissue. This transformation is accompanied by structural damage to tubular segments, fibrosis in the interstitium, and changes such as glomerular sclerosis [1]. It is a major contributing factor to the progressive impairment of kidney function [2].

MicroRNAs (miRNAs) constitute a group of short non-coding RNAs that, through complementary binding to the downstream target genes, can inhibit mRNA translation or facilitate mRNA degradation [3]. Numerous studies have confirmed the pivotal role of miRNAs in the fibrotic process [4,5]. However, research on the potential mechanisms of miR-10 in RF remains incomplete. The miR-10 family is a conserved group of microRNAs, including miR-10a and miR-10b. These two have been evolutionarily preserved within the Hox gene cluster, sharing the same seed sequence and playing similar roles in cellular activities. The literature confirms that the miR-10 family is predominantly expressed in the kidneys [6]. However, there is a lack of reported information on the potential mechanisms through which the miR-10 family may influence the key RF molecule Smad3.

Small Mothers Against Decapentaplegic Homolog 3 (Smad3) is a signaling transduction protein and a member of the Smad protein family. The Smad protein family plays a crucial role in regulating processes such as cell proliferation, differentiation, and apoptosis, serving as the principal regulators of signaling pathways like Transforming Growth Factor-β(TGF-β) and Bone Morphogenetic Protein (BMP) [7,8]. Phosphorylated Smad3 (p-Smad3), as a key influencer in fibrosis [9], modulates multiple signaling pathways, promoting the transformation of fibroblasts and collagen synthesis [10]. Consequently, this can exacerbate fibrotic reactions across various organ tissues, including the liver, lungs, and kidneys, among others. The existing literature strongly supports the upregulation of p-Smad3 expression in the UUO fibrosis model, intensifying fibrosis development [11].

Vasohibin-1 (VASH-1) is a protein known for its anti-angiogenic properties and is closely associated with various physiological processes, such as vascular development, tumor growth, tissue repair, and fibrosis [12,13,14,15,16]. Our preliminary research has demonstrated that the overexpression of the miR-10 family promotes the development of RF, with an additive effect observed for miR-10a and miR-10b [17]. Therefore, in this study, we focus on the combined impact of miR-10a and miR-10b. Given that VASH-1 serves as a common target gene for miR-10a and miR-10b, it is likely regulated by them, subsequently influencing downstream molecules involved in fibrosis. Recent studies have shed light on the roles of miR-10 and VASH-1 in the field of RF [17,18,19]. However, there are few reports on the impact of regulatory effects between the miR-10 family and VASH-1 resulting in RF.

Given these previous studies, this paper will delve into the roles of miR-10a and miR-10b in UUO RF mice and TGF-β1-induced HK-2 cells. Furthermore, we will explore the impact of the coexistence of miR-10a and miR-10b on the VASH-1/Smad3 pathway. Considering the current inadequacies in understanding the mechanisms of RF, our findings, by unraveling the interactions among these molecules, contribute to a deeper comprehension of the regulatory mechanisms underlying relevant biological processes. This, in turn, provides new targets and treatment strategies for the treatment of related diseases.

## 2. Results

### 2.1. Elevated Expression of miR-10a and miR-10b in UUO Mice

We assessed the expression of miR-10a and miR-10b in the UUO RF model (Figure 1A). A comparison was made among the sham-7d-WT, UUO-7d-WT, sham-14d-WT, and UUO-14d-WT groups. As expected, kidney edema and swelling, accompanied by a pale appearance, were observed after UUO surgery (Figure 1B). H&E staining revealed renal damage in the UUO group compared to the sham group, characterized by tubular dilation and interstitial fibrosis, with evident inflammatory cell infiltration. The degree of fibrosis significantly increased at 14 days. Masson staining results demonstrated a substantial presence of blue collagen protein in the kidneys of the model group, which intensified over time, indicating the success of our model (Figure 1C).

RT-qPCR results from homogenized mouse kidney tissues showed significant upregulation of miR-10a and miR-10b in the UUO group (Figure 1D). Protein imprint analysis of fibrosis-related protein levels in lysates yielded consistent conclusions (Figure 1E). Experimental results indicated a positive correlation between the severity of RF and the expression of miR-10a and miR-10b.

### 2.2. Depletion of miR-10a and miR-10b Alleviates Renal Fibrosis and Smad3 Phosphorylation Induced by UUO

To further validate the impact of the miR-10 family on RF, experiments were conducted using miR-10a^−/−^b^−/−^ (KO) mice (Figure 2A). Prior to modeling, the genotype of the mice was confirmed to ensure the absence of the miR-10a and miR-10b genes (Figure 2B). Subsequently, UUO surgery was administered, and kidneys were collected on day 7 and day 14. Analysis of H&E and Masson staining results revealed that compared to the sham group, UUO-injured kidneys exhibited features such as tubular atrophy and deformation, interstitial fibrosis, inflammatory cell infiltration, and occlusive lesions with the accumulation of blue collagen fibers (Figure 2C). After UUO surgery, the degree of fibrosis was reduced in the KO group compared to the WT group at the same time point. Additionally, we assessed changes in the fibrosis-related proteins Fibronectin, Col-III, and p-Smad3, a molecule closely involved in the development of RF. The results showed that, compared to the sham groups, the expression of Fibronectin, Col-III, and p-Smad3 increased after UUO surgery and continued to rise over time (Figure 2D). In the KO group, the expression of fibrosis-related proteins and p-Smad3 was lower than in the WT mice at the same time point. RT-qPCR experiments yielded consistent results (Figure 2E). Overall, the experimental data indicate that the absence of the miR-10 family can mitigate UUO-induced RF and alleviate Smad3 phosphorylation.

### 2.3. Overexpression of miR-10 Promotes TGF-β1-Induced Fibrosis and Smad3 Phosphorylation in HK-2 Cells

TGF-β1, known as a pro-fibrotic factor, triggers fibrosis-like changes in HK-2 cells. To validate the role of miR-10 in RF, we used TGF-β1 to induce HK-2 cells as a fibrosis model, selecting 20 ng/mL TGF-β1 based on previous studies [17]. Upon induction, cell morphology transitioned from a pavement stone-like state to an elongated and enlarged state (Figure 3A). With TGF-β1 stimulation, the expression of the fibrosis-related proteins Fibronectin and Col-III significantly increased, and p-Smad3 also showed a notable increase (Figure 3B). RT-qPCR experiments produced consistent results, and we observed a significant increase in the expression of miR-10a and miR-10b in HK-2 cells under TGF-β1 stimulation (Figure 3C).

Furthermore, we co-transfected miR-10a and miR-10b plasmids into HK-2 cells. Western blot results demonstrated that, under TGF-β1 stimulation, the co-transfection of miR-10a and miR-10b led to a significant increase in the expression of Fibronectin, Col-III, and p-Smad3. Conversely, the co-transfection of anti-miR-10a and anti-miR-10b attenuated the TGF-β1-induced increase in Fibronectin, Col-III, and p-Smad3 (Figure 3D). These findings imply that the miR-10 family can promote Smad3 phosphorylation and the progression of RF. The miR-10 family likely acts as a pivotal mediator in TGF-β1-induced RF and elevated p-Smad3 expression.

### 2.4. miR-10a and miR-10b Target and Downregulate VASH-1 Expression

The aforementioned experiments have confirmed that the miR-10 family promotes RF and Smad3 phosphorylation. Due to the high sequence similarity, we hypothesized that miR-10a and miR-10b might promote RF and Smad3 phosphorylation through the same mechanism. TargetScan was used to predict potential targets of miR-10a and miR-10b. The results revealed highly conserved base sequences in the 3′UTR region of VASH-1 mRNA between humans and mice, containing binding sites for highly conserved miR-10a-5p and miR-10b-5p.

To validate the binding of miR-10a and miR-10b to VASH-1, the predicted target 3′UTR region fragment was cloned downstream of the luciferase gene in the reporter plasmid, generating pcDNA3 EGFP/VASH-1-3′UTR-WT and pcDNA3 EGFP/VASH-1-3′UTR-Mut plasmids (Figure 4A). Validation was conducted through a luciferase reporter gene experiment. HEK 293T cells were co-transfected with miR-10a or miR-10b plasmids along with pcDNA3.0 EGFP, pcDNA3 EGFP/VASH-1-3′UTR-WT, and pcDNA3 EGFP/VASH-1-3′UTR-Mut. Western blot and EGFP fluorescence reporter gene experiment results demonstrated that the co-transfection of miR-10a or miR-10b with pcDNA3 EGFP/VASH-1-3′UTR-WT inhibited fluorescence from the 3′UTR, resulting in decreased EGFP protein expression (Figure 4B,C). This change was absent when the 3′UTR site was mutated.

When miR-10a and miR-10b were separately transfected into HK-2 cells; VASH-1 expression decreased at both the protein and mRNA levels (Figure 4D,E). Conversely, the transfection of anti-miR-10a or anti-miR-10b into HK-2 cells led to an increase in VASH-1 at both the protein and mRNA levels. These findings suggest that VASH-1 is a direct target of miR-10a and miR-10b, with a suppressive effect on its expression.

### 2.5. miR-10 Family Promotes Fibrosis by Modulating the VASH-1/Smad3 Pathway

To delve deeper into the mechanism underlying the miR-10 family’s involvement in RF, we conducted experiments to investigate VASH-1/Smad3 pathway modulation. Our findings confirmed a decrease in VASH-1 expression and an increase in p-Smad3 expression in UUO mice (Figure 2D and Figure 5A), which were consistent with the results from the TGF-β1-induced HK-2 cell model (Figure 5B), where p-Smad3 protein expression was notably elevated (Figure 3B).

To assess the functionality of VASH-1, we constructed VASH-1-related plasmids. After transfection with VASH-1, intracellular VASH-1 expression significantly increased (Figure 5C). Additionally, we constructed three knockdown plasmids, and through assessment, sh-VASH-1-3 had the best knockdown effect, so we used this plasmid in subsequent experiments (Figure 5D).

In TGF-β1-treated HK-2 cells, the upregulation of VASH-1 suppressed the expression of Fibronectin, Col-III, and p-Smad3. Conversely, inhibiting VASH-1 led to opposite results (Figure 5E). These findings indicate that elevated VASH-1 inhibits RF. Furthermore, to investigate the impact of the regulatory relationship between VASH-1 and the miR-10 family on RF, additional experiments were conducted to validate that overexpressing VASH-1 can attenuate the fibrosis-promoting effect of the miR-10 family (Figure 5F). This rescue experiment suggests that VASH-1 is regulated by the miR-10 family and is involved in the development of RF in HK-2 cells. Additionally, our experimental results demonstrate that TGF-β1 inhibits VASH-1 expression and promotes an increase in p-Smad3. When the miR-10 family is suppressed, it can weaken the TGF-β1-induced decrease in VASH-1 and the increase in p-Smad3. These results indicate that miR-10 can, at least partially, regulate the development of RF through the VASH-1/Smad3 pathway.

## 3. Discussion

In this study, we elucidated the involvement of miR-10 in the RF process and uncovered a potential signaling pathway, elucidating the role of the miR-10/VASH-1/Smad3 pro-fibrotic axis in TGF-β1-regulated RF. The validation of miR-10a and miR-10b roles was conducted through a unilateral ureteral obstruction (UUO) model in miR-10a^−/−^b^−/−^ mice. These results suggest that miR-10a and miR-10b may serve as novel targets for the treatment of RF (Figure 6).

Given the limited therapeutic approaches for RF and an incomplete understanding of its pathophysiology, there is a significant need to unravel the mechanisms driving its occurrence and progression [1,20]. The existing literature strongly supports the role of TGF-β1 in promoting fibroblast transformation, increasing matrix synthesis, inhibiting matrix degradation [21,22,23], and activating fibrotic pathways in the development of RF [24,25]. However, due to the complex signaling pathways in which TGF-β1 is involved, with diverse physiological functions and different regulatory effects on various cells and tissues [26,27,28], directly blocking TGF-β1 is not a feasible strategy. Hence, the emphasis should be on identifying more precise, fibrosis-related downstream pathways.

Smad3 serves as a crucial mediator in the TGF-β1 signaling pathway [29], exerting a pathogenic role in renal inflammation and fibrosis [30]. Based on our previous studies, it is evident that TGF-β1 possesses the capability to upregulate miR-10 expression. Through the modulation of miR-10, TGF-β1 can then regulate downstream target genes, contributing to the promotion of RF.

In our research group’s preliminary experiments, we observed an additive effect when miR-10a and miR-10b coexisted. Therefore, the focus of this study is to explore the combined role of miR-10a and miR-10b. MiR-10a and miR-10b belong to the miR-10 family and exhibit evolutionary conservation within the Hox developmental regulatory gene cluster across diverse species [31]. In mammals, miR-10a and miR-10b are located upstream of HoxB4 and HoxD4, respectively. Their highly similar sequences thermodynamically enable them to target the same mRNA [32]. Therefore, miR-10a and miR-10b may have similar functions, as confirmed in both our previous and current studies.

Through a combination of bioinformatics predictions, Western blot analysis, RT-qPCR, and EGFP fluorescent reporter assays, we have identified VASH-1 as a direct target of the miR-10 family. The miR-10 family can directly bind to the 3’UTR region of VASH-1, leading to the downregulation of its expression. VASH-1, an endogenous negative feedback regulator of angiogenesis, plays a role in regulating various factors to inhibit angiogenesis. It has been reported in diseases such as atherosclerosis, diabetic nephropathy, and pulmonary fibrosis [18,33,34]. Notably, studies have suggested that, apart from its association with the progressive decline in renal function [4], VASH-1 might potentially exert protective effects through negative feedback in renal diseases [18,35].

Building upon these findings, we investigated the role of the miR-10 family and VASH-1 in the development of RF. Our in vivo and in vitro experiments confirmed a decrease in VASH-1 expression, an increase in p-Smad3, and an elevation in the expression of fibrosis-related proteins in the model group. The overexpression of VASH-1 ameliorated these effects, highlighting its potential to alleviate Smad3 phosphorylation, thereby inhibiting the fibrotic pathway. It is noteworthy that although increasing VASH-1 expression partially mitigated the pro-fibrotic effects induced by miR-10, the expression levels of fibrosis-related proteins remained significantly higher than those in the control group. This implies that while VASH-1 is an important component in the TGF-β-induced RF process involving miR-10, it is not the sole regulatory factor by miR-10.

Currently, our research still faces some limitations. Further investigation is needed to fully elucidate the regulatory mechanism of TGF-β1 on miR-10. The process of fibrosis is intricate, involving multiple factors, molecules, and pathways, and different intervention methods exhibit significant variations in different RF models [36,37,38]. Therefore, no single model can completely replicate all aspects of kidney disease progression. It is crucial to validate the effects of miR-10 in different models and disease stages. Utilizing systems biology and bioinformatics approaches to integrate multi-omics data for establishing more comprehensive models is necessary. Additionally, exploring whether miR-10 exerts its regulatory effects through direct or indirect mechanisms with downstream molecules is vital for a thorough understanding of its role in kidney fibrosis. Investigating the regulatory mechanisms of the VASH-1/p-Smad3 pathway, including the molecules, signaling pathways, and regulatory modes involved, is also crucial for gaining a comprehensive understanding.

In summary, our study reveals an upregulation of the miR-10 family during the process of RF. This upregulation exacerbates the development of RF by promoting the phosphorylation of Smad3. Importantly, this mechanism is, at least partially, achieved through direct binding to the 3’UTR of VASH-1, leading to the suppression of VASH-1 expression. This mechanism may serve as a potential therapeutic strategy to modulate the progression of RF. It also suggests that miR-10 holds promise as a direction for treating RF. Further exploration of the feasibility and effectiveness of targeting the miR-10 family as a therapeutic approach for kidney diseases provides new insights into alleviating RF.

## 4. Materials and Methods

### 4.1. Gene Knockout Mice

Mice with knockout of the miR-10a and miR-10b genes were bred using CRISPR/Cas9 technology. BRL Pharmaceutical Company (Shanghai, China) provided the miR-10a knockout (miR-10a^−/−^) mice, while miR-10b knockout (miR-10b^−/−^) mice were obtained from WeishangLide Biotech Company (Beijing, China). By crossing miR-10a^−/−^ mice with miR-10b^−/−^ mice, we obtained miR-10a^−/−^10b^−/−^ (KO) mice. All these mice belong to the C57BL/6 genetic background. The gene typing of tail DNA samples was performed using agarose gel electrophoresis with the primers shown in Table 1.

### 4.2. Experimental Animal Grouping and Model Establishment

A total of 48 male C57BL/6 mice, eight weeks old, weighing 18 ± 2 g, were housed in an environment with a temperature of 22–24 °C, humidity of 50–60%, and a 12-h light/dark cycle, with free access to water and standard food. The protocol was approved by the Animal Ethics Committee of Tianjin Medical University (Doc. No. TMUaMEC 2022005). Prior to the experiments, a 2 mm tail segment was obtained for gene identification. In total, 24 wild-type (WT) mice and 24 knockout (KO) mice were randomly divided into eight groups (n = 6 each): (1) sham-7d-WT, (2) sham-7d-KO, (3) UUO-7d-WT, (4) UUO-7d-KO, (5) sham-14d-WT, (6) sham-14d-KO, (7) UUO-14d-WT, and (8) UUO-14d-KO. After anesthesia, the left ureter was exposed, ligated at the upper and lower thirds, and then cut. In the sham surgery group, mice underwent the same procedure except for ureter ligation. After surgery, mice were allowed free access to food and water. Kidneys were collected on day 7 or day 14 for further processing.

### 4.3. Western Blotting

Total protein from samples was extracted using RIPA buffer (Cwbio, Beijing, China). The protein concentration was then quantified using the BCA Protein Assay Kit (Cwbio, Beijing, China) and diluted accordingly. Equal amounts of protein were separated via SDS-PAGE with 8% or 10% polyacrylamide gels, depending on the molecular weight of the protein. The separated proteins were then transferred to a 0.45 μm PVDF membrane (Millipore,St. Louis, MO, USA) via electrophoresis, followed by blocking with 5% BSA at room temperature for 2 h. Primary antibodies for each protein were diluted in 5% BSA as follows: Fibronectin (1:1000, 26836S, CST), Col-III (1:1000, 66887S, CST), VASH-1 (1:1000, sc-365541, Santa Cruz), p-Smad3 (1:1000, 9520S, CST), Smad3 (1:1000, 9513S, CST), β-actin (1:1000, #AF7018, Affinity), and EGFP (1:5000, SRP15324, Saier); the secondary antibodies used were anti-rabbit (1:40,000, S0001, Affinity) and anti-mouse (1:40,000, S0002, Affinity). All primary antibodies were incubated overnight at 4 °C, followed by 1.5 h of incubation at room temperature with the corresponding secondary antibodies. Detection was performed using the ECL chemiluminescence detection kit (Biosharp, Guangzhou, China). Image collection and quantitative analysis were conducted using ImageJ software v1.8.0.

### 4.4. Real-Time Quantitative Polymerase Chain Reaction (RT-qPCR)

Total RNA from samples was extracted using Triquick Reagent (Solabio, Beijing, China). cDNA was synthesized from 1 μg of total RNA using HiScript^®^ II Q RT SuperMix (Vazyme, Nanjing, China), followed by quantitative real-time PCR with SYBR Green Master Mix (Abclonal, Wuhan, China). β-actin and snRNA U6 were utilized as internal control genes to normalize the relative expression levels of mRNA and miRNA. The cycle threshold (CT), at which point fluorescence first reached the preset threshold, was recorded to quantify the initial concentration of target gene mRNA and miRNA expression. Data were analyzed using the 2^−ΔΔCT^ method for statistical analysis. The primers used for RT-qPCR are as shown in Table 2.

### 4.5. Hematoxylin and Eosin (H&E) Staining and Masson’s Staining

After the embedding of renal tissues in paraffin, they underwent the deparaffinization process. Renal sections (4 μm) were stained with the H&E staining kit (Solarbio, Beijing, China). Images were captured at 100× magnification (Zeiss, Oberkochen, Germany). Fibrosis was indicated by renal tubular epithelial cells undergoing swelling, vacuolization, and exfoliation, along with disruption of the tubular structure. For Masson staining, renal sections (4 μm) were stained using the Masson staining kit (Solarbio, Beijing, China). Following UUO modeling, there was an increased deposition of collagen around the renal tubules, which appeared blue-stained.

### 4.6. Cell Culture and Transfection

HEK 293T cells and HK-2 cells (BNFUTURE, Beijing, China) were cultured at 37 °C with 5% CO_2_ in DMEM medium containing 10% fetal bovine serum (100991148, Gbico, Waltham, MA, USA) and 1% penicillin–streptomycin. All transfection experiments were performed using Lipofectamine TM 2000 transfection reagent (Invitrogen, Carlsbad, CA, USA) according to the manufacturer’s instructions. Prior to transfection, cells were seeded in 12-well plates, and transfection was initiated when the cell density reached 70% after 24 h. Total RNA was extracted 36 h post-transfection, and total protein was extracted 48 h post-transfection. For different treatments, equal volumes of PBS or 20 ng/mL TGF-β1 (MCE, Dallas, TX, USA) were added to the culture medium and continued to culture.

### 4.7. Plasmid Preparation and Oligonucleotide Synthesis

miRNA negative controls (PC and NC), as well as miR-10a, miR-10b, anti-miR-10a, and anti-miR-10b, were provided by the Basic Research Center of Tianjin Medical University. The human VASH-1 gene was cloned from the mixed DNA of normal epithelial cells, inserted into pcDNA3.1, and named pcDNA3.1-VASH-1 (VASH-1). To construct the VASH-1 knockdown plasmid, three sets of primers (sh-VASH-1-1, sh-VASH-1-2, sh-VASH-1-3) were designed, and they were connected to the NC vector through homologous recombination. The truncated element of the VASH-1 messenger RNA (mRNA) 3′UTR, containing the seed sequences for binding miR-10a and miR-10b, was amplified, annealed, and ligated downstream of the enhanced green fluorescent protein (EGFP) luciferase gene to construct the reporter vector (pcDNA3 EGFP-VASH-1-3′UTR-WT/Mut). All plasmids were confirmed for their effectiveness through sequencing and experimental validation. The primer sequences are provided in the table below (Table 3).

### 4.8. EGFP Fluorescent Reporter Assay

This analysis aimed to evaluate the targeting effects of miR-10a and miR-10b on VASH-1. HEK-293T cells were cultured in a 12-well plate, and after 24 h, when the cell density reached 70%, they were transfected with miR-10a, miR-10b, pcDNA3 EGFP, and pcDNA3 EGFP-VASH-1-3′UTR-WT/Mut plasmids. After 48 h, fluorescent brightness and quantity were observed under a fluorescence microscope, and cells were collected for Western blot analysis.

### 4.9. Statistical Analysis

The results were analyzed using GraphPad Prism 9.0 software and are presented as the mean ± standard deviation ($\overset{-}{x}$ ± SD) of three independent experiments, unless otherwise specified. Statistical significance between the two groups was determined using Student’s *t*-test, and a *p*-value < 0.05 was considered statistically significant. * *p* < 0.05, ** *p* < 0.01, *** *p* < 0.001, and **** *p* < 0.0001.

## 5. Conclusions

Both miR-10a/b can negatively regulate the target gene VASH-1, thereby promoting the phosphorylation of Smad3 and driving the development of renal fibrosis. miR-10a/b may serve as effective targets for treating renal fibrosis.

## Figures and Tables

**Figure 1 ijms-25-05232-f001:**
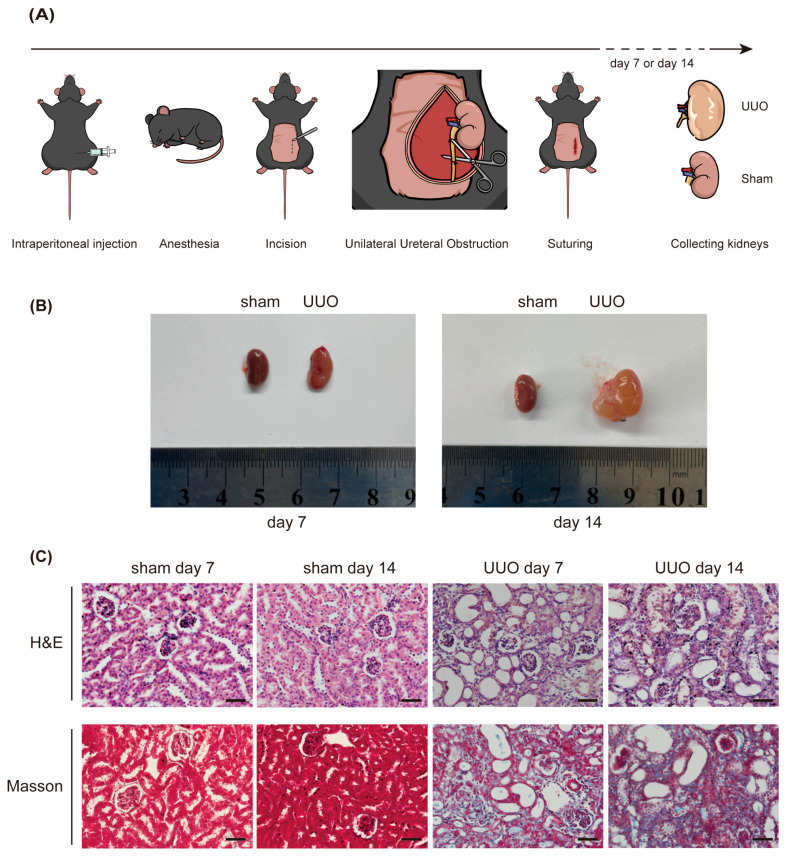
Elevated expression of miR-10a and miR-10b in UUO Mice. (**A**) Schematic diagram of UUO procedure. (**B**) Comparison images of the kidneys on day 7 and day 14 post-UUO modeling. (**C**) Representative images (superincumbent image scale bars, 50 µm) of kidney sections from the UUO model stained with H&E and Masson, illustrating day 7 and day 14; sham serves as the control. (**D**) RT-qPCR analysis of miR-10a and miR-10b in the UUO model, with snRNA U6 as the control. (**E**) Western blot analysis depicting the expression changes in fibrosis-related proteins in the kidneys of UUO model mice, with β-actin as the control. * *p* < 0.05, ** *p* < 0.01, *** *p* < 0.001, and **** *p* < 0.0001.

**Figure 2 ijms-25-05232-f002:**
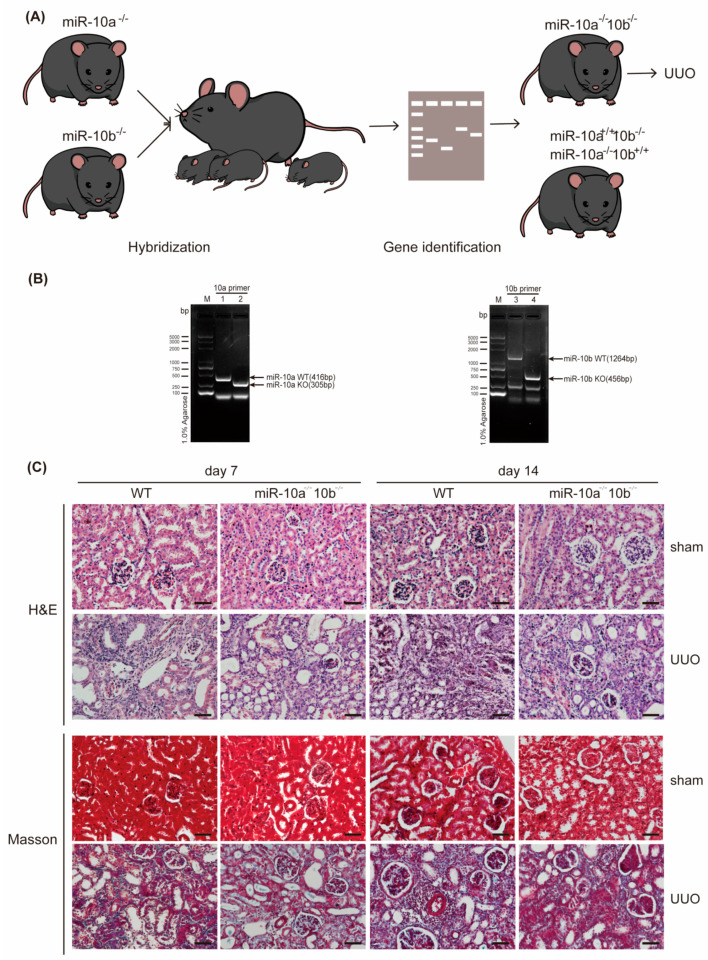
Depletion of miR-10a and miR-10b alleviates RF and Smad3 phosphorylation induced by UUO. (**A**) The process of obtaining KO mice through hybridization. (**B**) Genotype identification: comparison of miR-10a and miR-10b gene-deficient mice with WT mice. (**C**) Representative images (superincumbent image scale bars, 50 µm) of kidney sections from the UUO model stained with H&E and Masson, showing day 7 and day 14; sham serves as the control. (**D**) Expression results of Fibronectin, Col-III, and p-Smad3 in Western blot experiments, with β-actin as the control. (**E**) Expression results of Fibronectin and Col-III in RT-qPCR experiments, with β-actin as the control. * *p* < 0.05, ** *p* < 0.01, *** *p* < 0.001, and **** *p* < 0.0001.

**Figure 3 ijms-25-05232-f003:**
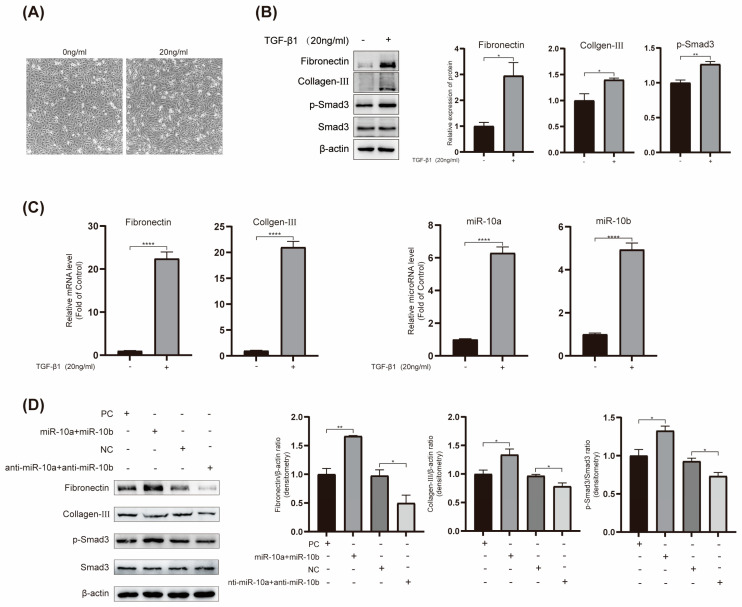
Overexpression of miR-10 promotes TGF-β1-induced fibrosis and Smad3 phosphorylation in HK-2 cells. (**A**) Morphological changes in HK-2 cells under 20 ng/mL TGF-β1 stimulation. (**B**) Western blot analysis of Fibronectin, Col-III, and p-Smad3 expression in HK-2 cells under TGF-β1 stimulation, with β-actin as a control. (**C**) RT-qPCR analysis of miR-10a, miR-10b, Fibronectin, and Col-III expression in HK-2 cells under TGF-β1 stimulation, with β-actin as a control. (**D**) Western blot examination of relevant target molecules in HK-2 cells transfected with the corresponding plasmids under TGF-β1 stimulation. * *p* < 0.05, ** *p* < 0.01, and **** *p* < 0.0001.

**Figure 4 ijms-25-05232-f004:**
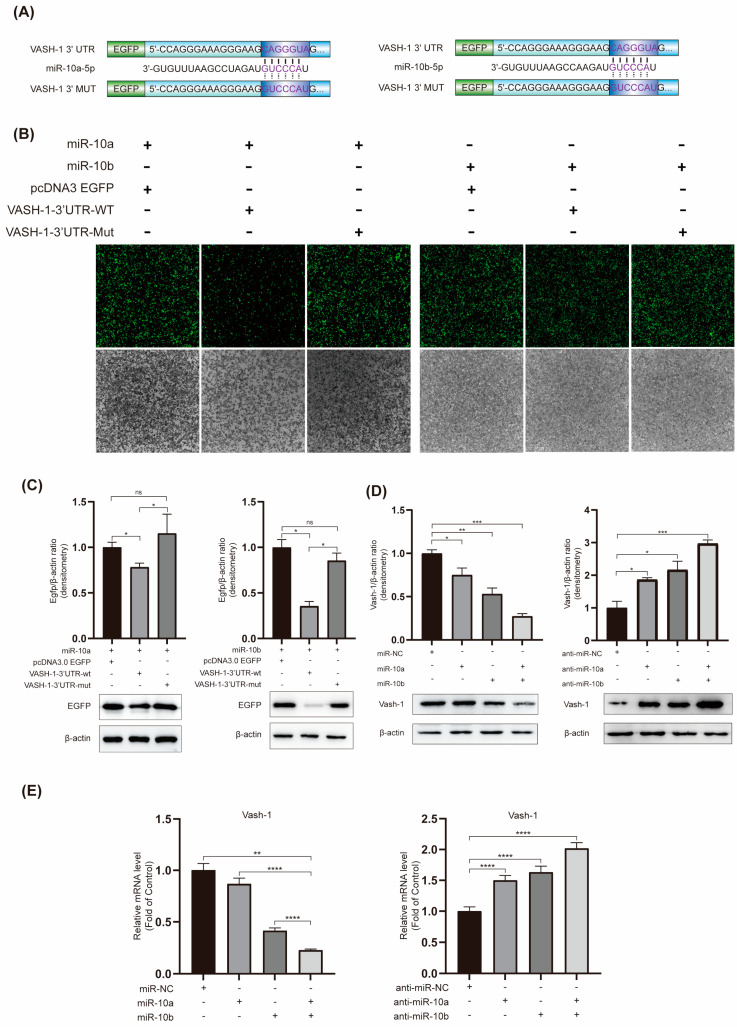
miR-10a and miR-10b target and downregulate VASH-1 expression. (**A**) Predicted binding sites of miR-10a and miR-10b with VASH-1 according to TargetScan, along with plasmid mutation sites. (**B**) EGFP fluorescence reporter gene experiment. (**C**) Western blot analysis of EGFP expression, with β-actin as a control. (**D**) Western blot analysis of the impact on VASH-1 expression after transfection with miR-10 family-related plasmids, with β-actin as a control. (**E**) RT-qPCR analysis of VASH-1 mRNA, with β-actin as a control. ns = not significant. * *p* < 0.05, ** *p* < 0.01, *** *p* < 0.001, and **** *p* < 0.0001.

**Figure 5 ijms-25-05232-f005:**
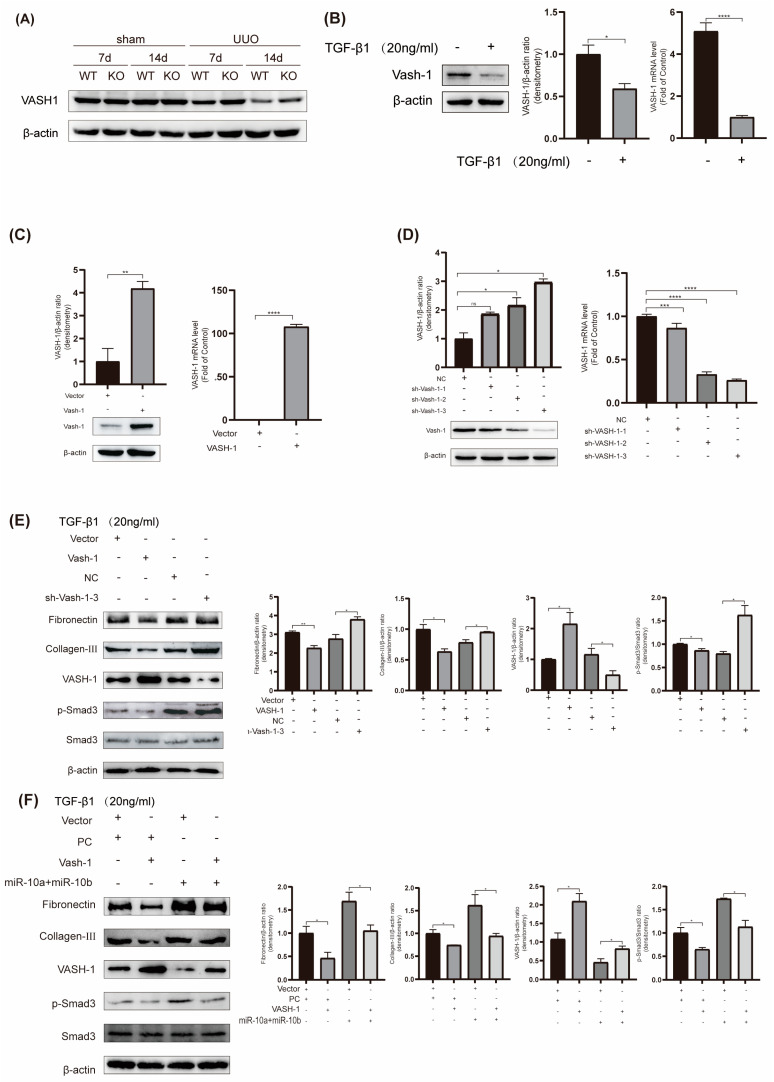
MiR-10 family promotes fibrosis by modulating the VASH-1/Smad3 pathway. (**A**) Western blot results of VASH-1 in mouse kidneys, with β-actin as a control. (**B**) After stimulation with 20 ng/mL TGF-β1, Western blot and RT-qPCR were performed to detect VASH-1 expression in HK-2 cells, with β-actin as a control. (**C**) After the transfection of HK-2 cells with overexpression plasmids, the results of Western blot and RT-qPCR experiments on the expression of VASH-1, with β-actin as a control, were determined. (**D**) After the transfection of HK-2 cells with knockdown plasmids, the results of Western blot and RT-qPCR experiments on the expression of VASH-1, with β-actin as a control, were determined. (**E**) Under stimulation with 20 ng/mL TGF-β1, after transfection with VASH-1-related plasmids, the protein expression results of Fibronectin, Col-III, and p-Smad3 in HK-2 cells, with β-actin as a control, were determined. (**F**) Under stimulation with 20 ng/mL TGF-β1, after transfection with relevant plasmids, the protein expression results of Fibronectin, Col-III, and p-Smad3 in HK-2 cells, with β-actin as a control, were determined. * *p* < 0.05, ** *p* < 0.01, *** *p* < 0.001, and **** *p* < 0.0001.

**Figure 6 ijms-25-05232-f006:**
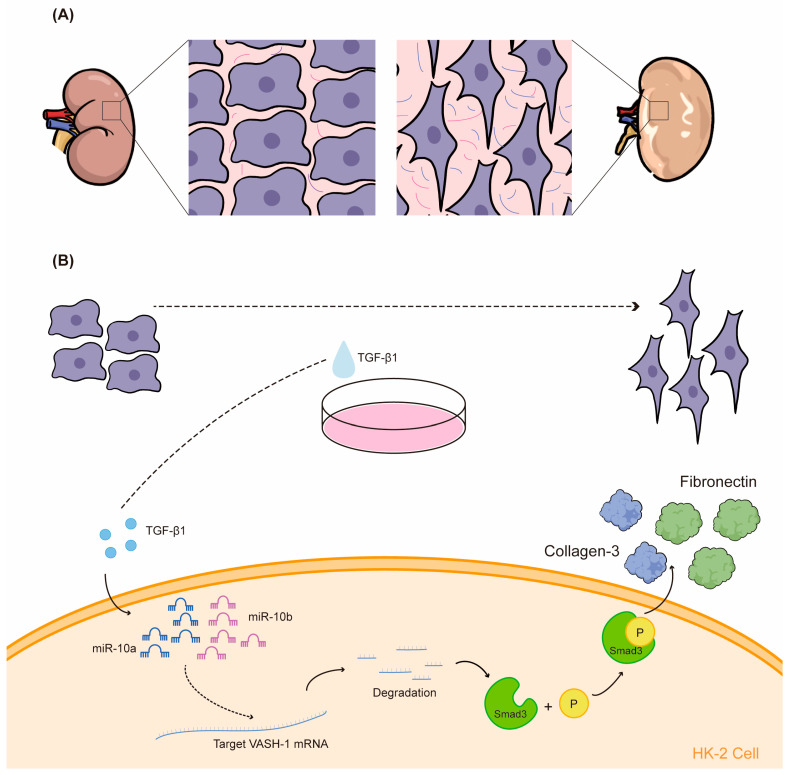
(**A**) Diagram illustrating changes after renal UUO surgery; (**B**) miR-10 family promotes RF development through the VASH-1/Smad3 pathway.

**Table 1 ijms-25-05232-t001:** Mouse gene identification primers.

Name	Primer
miR-10a-PCR-S	CCAAGAACGGACCCACAGT
miR-10a-PCR-A	AGTGAACAAGGACCCAAGC
miR-10b-PCR-F	CCAGAAAGGTAAATGCTCG
miR-10b-PCR-R	ATGAGTGTGGGCAATGTG

**Table 2 ijms-25-05232-t002:** Sequences of each primer.

Name	Primer
miR-10 RT	GTCGTATCCAGTGCAGGGTCCGAGGTGCACTGGATACGACAATTTGTG
U6 RT	GTCGTATCCAGTGCAGGGTCCGAGGTATTCGCACTGGATACGACAAAATATGGAAC
miR-10a-Forward	TGCGGTACCCTGTAGATCCGAATTTGTG
miR-10b-Forward	TGCGGTACCCTGTAGAACCGAATTTGTG
U6-Forward	TGCGGGTGCTCGCTTCGGCAGC
U6-Reverse	CCAGTGCAGGGTCCGAGGT
Fibronectin-F-human/mouse	GCTCAGCAAATCGTGCAGC
Fibronectin-R-human/mouse	CTAGGTAGGTCCGTTCCCACT
COL-III-F-human/mouse	CTGTAACATGGAAACTGGGGAAA
COL-III-R-human/mouse	CCATAGCTGAACTGAAAACCACC
VASH-1-F-human	GGTGGGCTACCTGTGGATG
VASH-1-R-human	CACTCGGTATGGGGATCTTGG
VASH-1-F-mouse	TGGGAATTTACCTCACCAACA
VASH-1-R-mouse	CCATAGGCCGCCTCATAG
β-actin-F-human/mouse	GGCTGTATTCCCCTCCATCG
β-actin-R-human/mouse	CCAGTTGGTAACAATGCCATGT

**Table 3 ijms-25-05232-t003:** The primers for the expression vectors.

Name	Primer
VASH-1-S	GTTAAGCTTGAATTCATGCCAGGGGGGAAG
VASH-1-A	CTTGTAATCCTCGAGGACCCGGATCTGGTA
shVASH-1-F	GATCCCCTGGGAATTTACCTCACCAACTCGAGTTGGTGAGGTAAATTCCCAGGTTTTTGA
shVASH-1-R	AGCTTCAAAAACCTGGGAATTTACCTCACCAACTCGAGTTGGTGAGGTAAATTCCCAGGG
shVASH-2-F	GATCCCTGCCAATCAAATGCCTGGAACTCGAGTTCCAGGCATTTGATTGGCAGTTTTTGA
shVASH-2-R	AGCTTCAAAAACTGCCAATCAAATGCCTGGAACTCGAGTTCCAGGCATTTGATTGGCAGG
shVASH-3-F	GATCCCCTACTTCTCAGGGAACTACTCTCGAGAGTAGTTCCCTGAGAAGTAGGTTTTTGA
shVASH-3-R	AGCTTCAAAAACCTACTTCTCAGGGAACTACTCTCGAGAGTAGTTCCCTGAGAAGTAGGG
pcDNA3 EGFP-VASH-1-3′UTR-WT-F	AGCTGTACAAGTAAAGGATCCCCAGGGAAAGGGAAGCAGGGTAGGAATTCTGCAGATATCCAGCA
pcDNA3 EGFP-VASH-1-3′UTR-WT-R	TGCTGGATATCTGCAGAATTCCTACCCTGCTTCCCTTTCCCTGGGGATCCTTTACTTGTACAGCT
pcDNA3 EGFP-VASH-1-3′UTR-MUT-F	AGCTGTACAAGTAAAGGATCCCCAGGGAAAGGGAAGGTCCCATGGAATTCTGCAGATATCCAGCA
pcDNA3 EGFP-VASH-1-3′UTR-MUT-R	TGCTGGATATCTGCAGAATTCCATGGGACCTTCCCTTTCCCTGGGGATCCTTTACTTGTACAGCT

## Data Availability

Data are contained within the article.

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
