# Peer review of "MicroRNA-10 Family Promotes Renal Fibrosis through the VASH-1/Smad3 Pathway"

_ijms, 2024, doi:10.3390/ijms25105232_

Round 1
Reviewer 1 Report
Comments and Suggestions for Authors
This study needs major revision to clarify/corerct the following points:
Line 31: better ‘tubular segments’ or ‘tubules’ than ‘renal units’. Every other nephron segment is mentioned below, so it would be appropriate to mention one of the most important ones.
Figure 1A: In the 6th photomicrograph, the UUO kidney could be depicted as larger than the contralateral, as there is a visible increase in the kidney size (when filled with urine), and this can also be observed in Figure 1B.
Figure 1C and 2C: with this size of the photomicrographs used, the magnification should be higher. Although the damage is mostly visible, it is difficult to look at, especially in Figure 2C, also the resolution/image quality should be improved.
Line 80: ‘dull appearance’ is not a commonly used term to describe the kidney appearance. What exactly did the authors mean here?
Line 82: ‘mesangial expansion’ is not a typical sign in the UUO model. How was this quantified? Atubular glomeruli and some sclerotic glomeruli are common, but ‘mesangial expansion’ is typical for some specific glomeruli-originating diseases.
Line 82: how was ‘interstitial deformation’ defined? For this type of injury it is typical to see tubular dilation, tubular atrophy, interstitial fibrosis and infiltration of inflammatory cells - I assume that the latter corresponds to ‘local inflammatory reactions’?
Line 83-84: How are ‘renal units’ defined here?
Line 88: ‘Qrt-PCR’ is not a commonly used abbreviation for the technique.
Figure 1E: Collagen III is the correct use for protein depiction (not Collagen 3). The x-axis labels of the Fibronectin and Collagen state ‘shame’ instead of ‘sham’. ‘Shame’ has a completely different meaning…
Line 108: What kind of ‘vascular injury’?
Line 113: fibronectin in capital letters
Figure 2 - A,B,C,D,E should be the order of the photomicrographs.
Line 132 - again, higher magnification has to be provided to see the changes mentioned.
Figure 4B, E - higher magnification has to be provided to see the changes mentioned.
Some minor use of language, while grammatically correct, is not used properly, e.g.: 'UUO treatment was administered'.
Author Response
Dear Reviewer,
Thank you for reviewing our manuscript and providing valuable suggestions and feedback. We truly appreciate your attention and input into our research.
We have carefully reviewed and considered the suggestions you provided. We will make revisions to the manuscript according to your recommendations to improve its quality and comprehensibility. Below is our response to the issues and comments you raised:
Line 31: better ‘tubular segments’ or ‘tubules’ than ‘renal units’. Every other nephron segment is mentioned below, so it would be appropriate to mention one of the most important ones.
Thank you for your suggestion. We have changed it to "renal tubules" as per your advice, indeed emphasizing this aspect would be more appropriate.
Figure 1A: In the 6th photomicrograph, the UUO kidney could be depicted as larger than the contralateral, as there is a visible increase in the kidney size (when filled with urine), and this can also be observed in Figure 1B.
Thank you for your suggestion. We have made the modification in Figure 1A as per your advice, replacing it with a more representative image. Originally, Figure 1A aimed to depict the kidneys after urine voiding; however, aligning it with Figure 1B makes more sense. We have replaced it with a schematic illustration of the kidneys swelling after being filled with urine.
Figure 1C and 2C: with this size of the photomicrographs used, the magnification should be higher. Although the damage is mostly visible, it is difficult to look at, especially in Figure 2C, also the resolution/image quality should be improved.
Thank you for your suggestion. We have replaced the photo with one at 200x magnification and improved the resolution as per your advice.
Line 80: ‘dull appearance’ is not a commonly used term to describe the kidney appearance. What exactly did the authors mean here?
Thank you for your suggestion. We intended to convey that the kidneys became "pale and dull" in color, but we have temporarily removed "dull" as we haven't found a more suitable replacement term yet. If you have any better suggestions, please let me know, and I would greatly appreciate it.
Line 82: ‘mesangial expansion’ is not a typical sign in the UUO model. How was this quantified? Atubular glomeruli and some sclerotic glomeruli are common, but ‘mesangial expansion’ is typical for some specific glomeruli-originating diseases.
Thank you for your suggestion. We intended to express the tubular dilation caused by UUO. We appreciate your rigorous approach in pointing out our shortcomings, and we will carefully verify the relevant information to ensure accuracy.
Line 82: how was ‘interstitial deformation’ defined? For this type of injury it is typical to see tubular dilation, tubular atrophy, interstitial fibrosis and infiltration of inflammatory cells - I assume that the latter corresponds to ‘local inflammatory reactions’?
Thank you for your suggestion. We believe that UUO leads to tubular obstruction, resulting in tubular dilation, tubular atrophy, and interstitial fibrosis. This tubular obstruction triggers a series of pathological changes, including local inflammatory reactions, interstitial cell proliferation, and interstitial fibrosis. We initially referred to these changes as "interstitial deformation." However, considering that the term "interstitial deformation" might not be appropriate, we have revised it to "interstitial fibrosis."
Line 83-84: How are ‘renal units’ defined here?
Thank you for your suggestion. What we intended to express was "tubular deformation." As some of the expressions in this paragraph overlap with the previous text, we have removed them accordingly.
Line 88: ‘Qrt-PCR’ is not a commonly used abbreviation for the technique.
Thank you for your suggestion. What we meant to convey with "Qrt-PCR" is "Quantitative reverse transcription polymerase chain reaction," which we will change to "RT-qPCR."
Figure 1E: Collagen III is the correct use for protein depiction (not Collagen 3). The x-axis labels of the Fibronectin and Collagen state ‘shame’ instead of ‘sham’. ‘Shame’ has a completely different meaning…
Thank you for your suggestion. We will change it to "Collagen III". "Sham" was a typographical error, and we will correct it.
Line 108: What kind of ‘vascular injury’?
In this section, we intended to convey "vascular lumen compression or dilation". However, since we replaced pathological images, this change may not be evident in the figures. Therefore, we will remove this statement here.
Line 113: fibronectin in capital letters
Thank you for your suggestion. We have made modifications to the writing.
Figure 2 - A,B,C,D,E should be the order of the photomicrographs.
Thank you for your suggestion. We have made adjustments to the sequence of images.
Line 132 - again, higher magnification has to be provided to see the changes mentioned.
Thank you for your suggestion. We have enlarged the photos in the hope that this change will meet the requirements.
Figure 4B, E - higher magnification has to be provided to see the changes mentioned.
Thank you for your suggestion. Regarding 4B: Since it was necessary to observe the quantity and brightness of fluorescence across the entire field of view, a lower magnification was chosen to observe a larger area. Therefore, we did not replace the image, but increased its size to make the changes more visible, hoping this adjustment is acceptable. Regarding 4E: We have enlarged the image in the hope that this change will meet the requirements.
Some minor use of language, while grammatically correct, is not used properly, e.g.: 'UUO treatment was administered'.
Thank you for your suggestion. We have revised the wording to " UUO surgery."
Thank you once again for your review and feedback. We look forward to your re-evaluation of the revised manuscript and are eager to work together with you to elevate the paper to a higher standard.
Warm regards,
Sincerely yours,
Guoxia Li
Reviewer 2 Report
Comments and Suggestions for Authors
In the investigation led by Yichen Shuai et al., "MicroRNA-10 family promotes renal fibrosis through the VASH-1/Smad3 pathway," the authors effectively employed in vivo and in vitro models to investigate the role of miR-10a and miR-10b in renal fibrosis (RF). Utilizing a unilateral ureteral obstruction (UUO) model in mice and TGF-β1-stimulated HK-2 cells, they observed elevated miR-10a/b expression and its possible effects on VASH-1 and p-Smad3.
Major comment:
There appears to be a significant discrepancy between the statements regarding the effect of miR-10a and miR-10b on p-Smad3 expression in the text and the data presented in Figure 3D. While the text suggests that "co-transfection of anti-miR-10a and anti-miR-10b attenuated the TGF-β1-induced increase in Fibronectin, Col-3, and p-Smad3," the Western blot image indicates a decrease in fibronectin and collagen 3 but an increase in the expression of p-Smad3. This inconsistency raises concerns about the accuracy and reliability of the conclusions drawn by the authors. Ensuring that the text accurately reflects the findings presented in the Western blot image and densitometry data is crucial for maintaining the integrity and credibility of the research. It is strongly advised that the authors carefully review their data and provide a clear explanation for this discrepancy to uphold transparency and scientific rigor in their analysis.
Minor comments:
- In paragraph four of the introduction, on line 63, the statement "Recent studies have shed light on the roles of miR-10 and VASH-1 in the field of RF" requires a reference to support its claim.
- In line 79, it is expected that following UUO surgery, the affected kidney would develop hydronephrosis; therefore, using "as expected" instead of "noticeable" would be more appropriate.
- In lines 83-84, please clarify the term "renal units" for better understanding.
- In line 98, it would be helpful to clarify what "U6" stands for in the legend.
- HE contraction for Hematoxylin and eosin staining is ambiguous, as 'HE' could be mistaken for the English pronoun. Consider expanding 'HE' to 'H&E' for clarity and consistency with the standard abbreviation used for this staining technique.
- In line 107, consider using "UUO surgery" or "UUO-injured kidney" instead of "UUO-treated" since UUO is a surgical procedure, not a therapy.
- In line 113, please use lowercase for "FIBRONECTIN" in this and subsequent lines for consistency.
- In line 118, consider the term "alleviate" when expressing a decrease in the expression of the phosphorylated form of Smad3.
- In line 131, the phrase "selecting 20ng/ml TGF-β1 based on previous studies" requires a reference to support its validity.
Author Response
Dear Reviewer,
Thank you for reviewing our manuscript and providing valuable suggestions and feedback. We truly appreciate your attention and input into our research.
We have carefully reviewed and considered the suggestions you provided. We will make revisions to the manuscript according to your recommendations to improve its quality and comprehensibility. Below is our response to the issues and comments you raised:
Major comment:
There appears to be a significant discrepancy between the statements regarding the effect of miR-10a and miR-10b on p-Smad3 expression in the text and the data presented in Figure 3D. While the text suggests that "co-transfection of anti-miR-10a and anti-miR-10b attenuated the TGF-β1-induced increase in Fibronectin, Col-3, and p-Smad3," the Western blot image indicates a decrease in fibronectin and collagen 3 but an increase in the expression of p-Smad3. This inconsistency raises concerns about the accuracy and reliability of the conclusions drawn by the authors. Ensuring that the text accurately reflects the findings presented in the Western blot image and densitometry data is crucial for maintaining the integrity and credibility of the research. It is strongly advised that the authors carefully review their data and provide a clear explanation for this discrepancy to uphold transparency and scientific rigor in their analysis.
Thank you for your scientific and rigorous approach in questioning and providing suggestions on the issues raised in the article. Upon careful examination, we have identified that an incorrect WB image was used in Figure 3D due to oversight. However, the original images submitted with the initial article submission were correct. We will rectify the image and upload the corrected version. Moving forward, we will conduct thorough checks on our article with greater diligence. We appreciate your corrections and critiques.
Minor comments:
In paragraph four of the introduction, on line 63, the statement "Recent studies have shed light on the roles of miR-10 and VASH-1 in the field of RF" requires a reference to support its claim.
Thank you for your suggestion. We have included the relevant references in the citations.
In line 79, it is expected that following UUO surgery, the affected kidney would develop hydronephrosis; therefore, using "as expected" instead of "noticeable" would be more appropriate.
Thank you for your suggestion. We have revised the wording to make it more appropriate.
In lines 83-84, please clarify the term "renal units" for better understanding.
Thank you for your suggestion. We believe that the description of "renal units" here lacks specificity, and we intend to refer to the changes in renal tubules. However, considering the redundancy in the expression before and after, we have deleted the redundant descriptions here.
In line 98, it would be helpful to clarify what "U6" stands for in the legend.
Thank you for your suggestion. What we intended to express is "housekeeping gene U6 in the control group," which is a type of small nuclear RNA used as an internal reference gene in our study.
HE contraction for Hematoxylin and eosin staining is ambiguous, as 'HE' could be mistaken for the English pronoun. Consider expanding 'HE' to 'H&E' for clarity and consistency with the standard abbreviation used for this staining technique.
Thank you for your suggestion. We have changed "HE" to "H&E" to avoid ambiguity.
In line 107, consider using "UUO surgery" or "UUO-injured kidney" instead of "UUO-treated" since UUO is a surgical procedure, not a therapy.
Thank you for your suggestion. We have made modifications to this expression.
In line 113, please use lowercase for "FIBRONECTIN" in this and subsequent lines for consistency.
Thank you for your suggestion. We have made modifications here.
In line 118, consider the term "alleviate" when expressing a decrease in the expression of the phosphorylated form of Smad3.
Thank you for your suggestion. We have made modifications to the corresponding sections.
In line 131, the phrase "selecting 20ng/ml TGF-β1 based on previous studies" requires a reference to support its validity.
Thank you for your suggestion. We have added relevant citations accordingly.
Thank you once again for your review and feedback. We look forward to your re-evaluation of the revised manuscript and are eager to work together with you to elevate the paper to a higher standard.
Warm regards,
Sincerely yours,
Guoxia Li
Round 2
Reviewer 1 Report
Comments and Suggestions for Authors
estions, please let me know, and I would greatly appreciate it.
Response #2: Here are the synonyms for the word ‘dull’ using the Thesaurus Dictionary (https://www.thesaurus.com/browse/dull): e.g.: unintelligent, boring, dim, dumb, simple, slow, sluggish etc… I was simply asking which of these characteristics did the authors had in mind when describing an UUO kidney.
Ad Figure 4B: The quality is still somewhat poor. I suggest this is ran by the journal’s image quality checkers.
Further minor issues:
It is more common to use ‘day 7’ or ‘day 14’ instead of ‘7 day’ or ‘14 day’.
Table 1: What is ‘miR-10a-PCR-S’ and ‘miR-10a-PCR-A’ ?
Comments on the Quality of English Language
Minor editing of English language required
Author Response
Dear Reviewer,
Thank you once again for your thorough and diligent review of our manuscript and for providing valuable feedback. We have carefully considered all of your suggestions and comments, and we would like to address them as follows:
Response #2: Here are the synonyms for the word ‘dull’ using the Thesaurus Dictionary (https://www.thesaurus.com/browse/dull): e.g.: unintelligent, boring, dim, dumb, simple, slow, sluggish etc… I was simply asking which of these characteristics did the authors had in mind when describing an UUO kidney.
We greatly appreciate your assistance and suggestions, and the Thesaurus Dictionary was very helpful. What we want to describe here is that the kidneys after the UUO surgery appear lighter in color compared to normal kidneys and don't look as vibrant. We are now using the word "pale" to describe this condition.
Ad Figure 4B: The quality is still somewhat poor. I suggest this is ran by the journal’s image quality checkers.
Thank you for your suggestion. We aim to depict the inhibitory effects of miR-10a and miR-10b on downstream target genes through these images. This effect is reflected not only in fluorescence intensity but also in the quantity of fluorescence. Therefore, initially, we chose a lower magnification to better observe the overall fluorescence effect. In this revision, we have replaced the images with higher magnification based on your suggestion.
Further minor issues:
It is more common to use ‘day 7’ or ‘day 14’ instead of ‘7 day’ or ‘14 day’.
Thank you for your suggestion. We have made the necessary modifications accordingly.
Table 1: What is ‘miR-10a-PCR-S’ and ‘miR-10a-PCR-A’ ?
Thank you for your inquiry. Here, 'miR-10a-PCR-S' and 'miR-10a-PCR-A' are names of primer sets, with 'A' and 'S' representing "antisense" and "sense," respectively, indicating the antisense primer and sense primer. Since the names of miR-10a and miR-10b are very similar and their primer sequences are also highly similar, we used 'A' and 'S', 'F' and 'R' to distinguish between these two pairs of primers. However, 'A' and 'S', 'F' and 'R' have no practical difference in meaning. We hope this explanation addresses your concerns.
We are grateful for the opportunity to improve our work with your guidance.
Thank you for your time and consideration.
Best regards,
Guoxia Li
Reviewer 2 Report
Comments and Suggestions for Authors
Thank you for your thorough response to my comments. I appreciate your attention to detail in addressing both major and minor concerns. It's reassuring to know that you've identified the issue with the incorrect Western blot image and are taking steps to rectify it. Your commitment to upholding transparency and scientific rigor is commendable.
Regarding the minor comments, I'm glad to see that you've made the necessary revisions to improve clarity and consistency throughout the manuscript. The inclusion of references where needed enhances the credibility of your claims, and the adjustments in terminology and formatting contribute to the overall professionalism of the paper.
Author Response
Dear Reviewer,
Thank you once again for your thorough and diligent review of our manuscript and for providing valuable feedback. We greatly appreciate your correction of the errors in our manuscript. A rigorous and transparent research environment is a goal we all strive for. Thank you for acknowledging our efforts in making the necessary revisions. Your critique and suggestions have been invaluable to us, and we will endeavor to further refine our manuscript to enhance its professionalism.
We are grateful for the opportunity to improve our work with your guidance.
Thank you for your time and consideration.
Best regards,
Guoxia Li
Round 3
Reviewer 1 Report
Comments and Suggestions for Authors
Thank you for answering all the questions, I have no further comments.
Comments on the Quality of English LanguageMinor editing of English language required